

# Surface electromyographic analysis of differential effects in kettlebell carries for the serratus anterior muscles

Alex Caravan[1],*, John O. Scheffey[1],*, Sam J. Briend[2] and Kyle J. Boddy[1]

[1] Research and Development, Driveline Baseball, Inc, Kent, WA, USA
[2] High Performance, Driveline Baseball, Inc, Kent, WA, USA
* These authors contributed equally to this work.

## ABSTRACT

The purpose of this study was to examine differences in the Electromyography (EMG) amplitude of the serratus anterior between 45° kettlebell carries and 90° kettlebell carries. Thirty-three men aged roughly between 19 and 23 and who were either college or professional baseball pitchers were chosen and randomly assigned to either perform the 45° kettlebell carry followed by the 90° kettlebell carry ($n = 17$) or the 90° kettlebell carry followed by the 45° kettlebell carry ($n = 16$). Each pitcher was instructed in the proper usage of the exercise and assigned a short break between the two carries. Changes in EMG amplitude were examined after proper band-pass filtering, normalization, and moving average-smoothing of the raw EMG signal. Differences of the EMG amplitude mean frequencies were examined between each subject's individual carries and the clumped groups of all 45° and 90° carries. Among each individual comparison, eight pitchers had "large" Effect Size differences between the EMG amplitudes of their two carries, with seven of them signaling the 45° carry as the larger value. In addition, when examining the grouped mean differences of the EMG amplitudes, we found the 45° carries to be significantly higher ($p$-value of 0.018).

## INTRODUCTION

For overhead athletes poor scapula function can result in a myriad of issues, the least of which are adverse force generation and a propensity for shoulder injury (*Myers et al., 2005*). To counter these detrimental effects for overhead athletes' bottoms-up kettlebell carries, or kettlebell carries, are commonly used as a recovery, rehabilitation and strength training exercise. Activation of the serratus anterior is the intended focus of these exercises as it is a prime mover for both scapular upward rotation and scapula stabilization, especially during overhead movements (*Phadke, Camargo & Ludewig, 2009*). In turn, these movements help limit subacromial impingement and stabilize the scapula (*Ludewig & Cook, 2000*).

However, to our knowledge, no studies exist that examine muscle activation of overhead stabilizers like the serratus anterior during kettlebell carries. Several variations

Corresponding author
Kyle J. Boddy,
kyle@drivelinebaseball.com

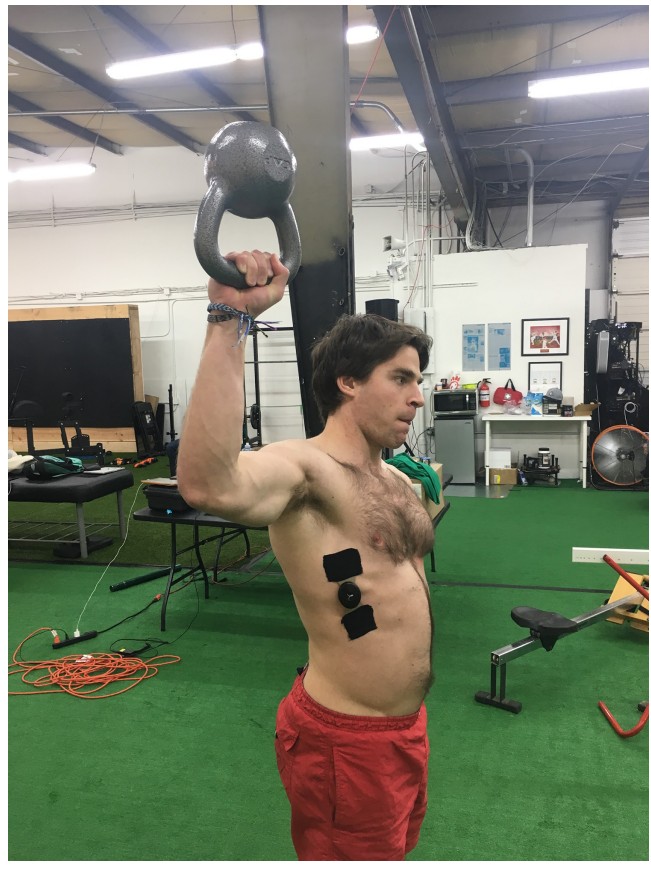

**Figure 1 Kettlebell carry at 90°.** An example of a subject carrying the kettlebell at approximately 90° of horizontal shoulder abduction. Photo credit: John O. Scheffey.

of the kettlebell carry are commonly used, with little data comparing the effectiveness of different variations.

This study involved measurements of electromyography (EMG) amplitude to examine differences across various positions and exercises, following similar precedent from past studies (*Decker et al., 1999*). EMG is commonly used to measure electric potential differences caused by muscle activation. However, because it can be affected by various forms of noise, including equipment calibration, background electromagnetic radiation, movement of electrodes, randomness of muscle firing patterns, and electrocardiography signal, certain to-be-discussed analytical methods were undertaken to maximize the bifurcation of the signal and the noise including the filtering, normalization, and smoothing of the raw EMG signal.

The purpose of this study was to use EMG sensors to examine serratus anterior activation in baseball pitchers performing bottoms up kettlebell carries at 90° and *approximately* 45° of horizontal abduction of the humerus, and to determine if there is a statistically significant difference in the EMG amplitude of the serratus anterior between the two variations. Zero degrees was set as the arm being extended in front of the person, and 90° was set as the arm being extended out to the side (Figs. 1 and 2).

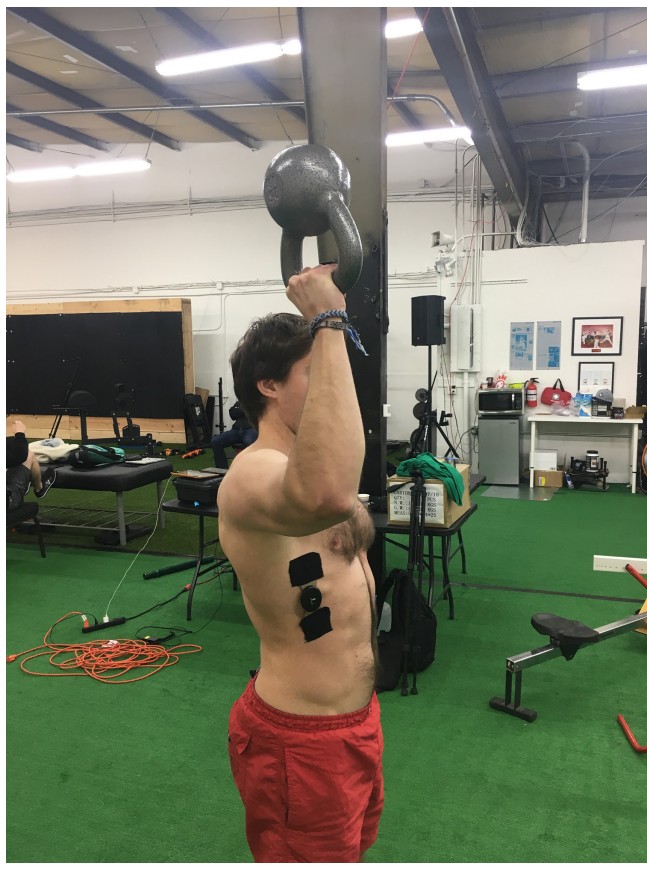

**Figure 2 KB carry at 45°.** An example of a subject carrying the kettlebell at approximately 45° of horizontal shoulder abduction. Photo credit: John O. Scheffey.

Kettlebell carry variations that cause a higher serratus anterior amplitude may imply higher muscle activation and may be more beneficial for the recovery and rehabilitation of overhead throwing athletes.

## METHODS

### Subjects

The study was approved by Hummingbird IRB, which granted ethical approval to carry out the study (Hummingbird IRB# 2017-58) at the author's facilities. Subjects were provided a verbal explanation of the study and then given an informed consent form to read and sign before participating. Forty healthy college and professional pitchers training at the Driveline Baseball facility in Kent, Washington were selected as participants in the study. Due to failure to follow instructions and mismatched data this number was reduced, with 33 total subjects having their data included in the final analysis (age = 22.2 ± 3.5 years; height = 73.4 ± 2.1 inches; body weight = 200.8 ± 17.6 lbs). Inclusion criteria required subjects to be between 18 and 40 years of age and not possess a significant medical history of shoulder or rotator cuff injuries. Subjects were required to be able to lift at least 50 lbs from the ground without discomfort or pain in the upper

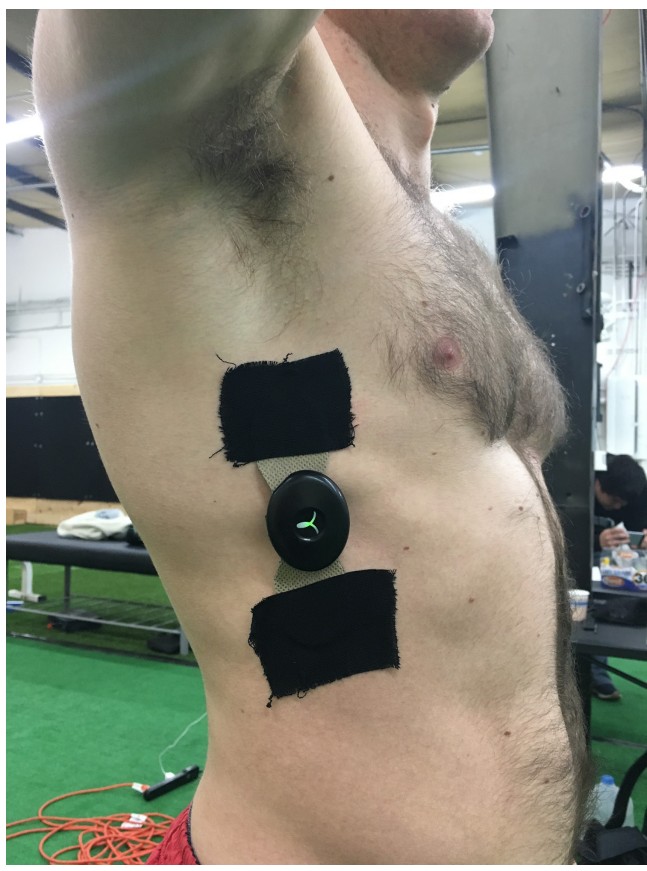

**Figure 3 EMG application example.** Application of the Somaxis EMG sensor to the subject. Photo credit: John O. Scheffey.

extremities, specifically the forearm, elbow, shoulder, and neck areas. If pain or discomfort was experienced at any point during trials, the study was stopped and the participant's data were not included.

## Procedures

Subjects were recruited to participate in the study after throwing, as part of their post-training recovery exercises. The exercise is intended to serve as a post-throwing recovery exercise in order to reduce risk of arm pain and instability. Once ready to participate, subjects were selected from the general population and placed in the two distinct groups of 20 subjects through a block randomization process. The serratus anterior of the dominant arm was cleaned and a Somaxis Cricket EMG sensor (Somaxis Inc., San Francisco, CA, USA) was applied vertically across a standardized site on the serratus anterior (Fig. 3). The EMG sensor was held in place using kinesiology tape (RockTape, Inc., Campbell, CA, USA).

Prior to performing kettlebell carries, maximum voluntary isometric contraction (MVIC) data for each subject was collected, to normalize future EMG readings for the kettlebell carries. Subjects were instructed to raise their dominant arm to 125° of shoulder abduction in the scapular plane (not to be confused with the previous references to the horizontal humeral plane) and a goniometer was used to measure the exact range of

10.7717/peerj.5044

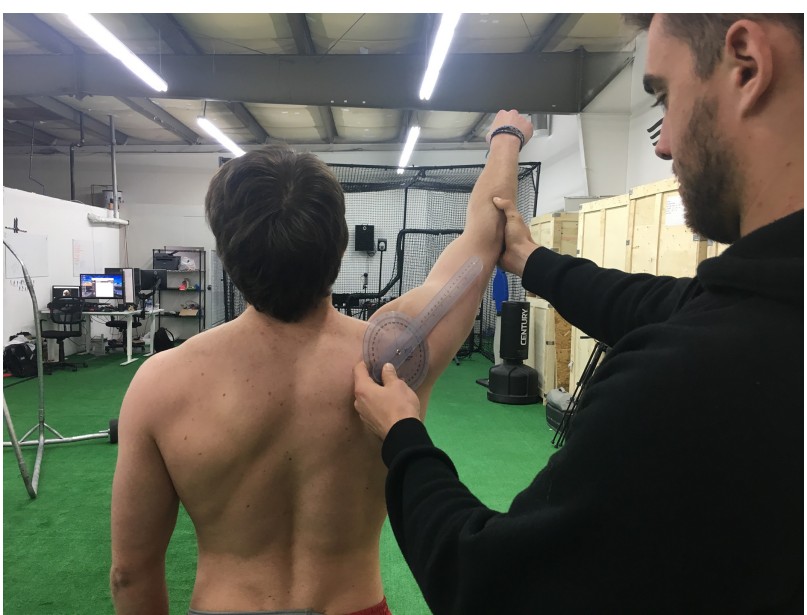

**Figure 4 MVIC test 1.** The first MVIC test done on subjects. Photo credit: John O. Scheffey.

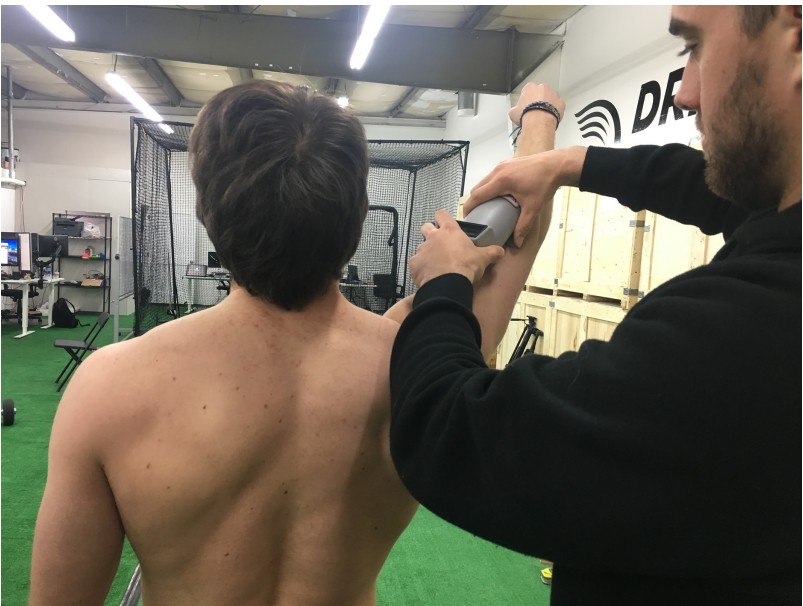

**Figure 5 MVIC test 2.** The second image in the MVIC test series. Photo credit: John O. Scheffey.

motion. Subjects were then instructed, with their arm raised in the same measured position, to resist a downward force of 5 lbs applied to the humerus with a hand-held dynamometer (Lafayette Instrument Co, Lafayette, IN, USA) by an experimenter, to achieve MVIC (Figs. 4 and 5). Manual resistance at a shoulder abduction angle of about 120° has been shown to achieve higher serratus anterior activation than other MVIC positions (*Jung & Moon, 2015*).

Subjects were then coached in the proper technique for kettlebell carries with a standardized set of instructions. The first group performed a bottoms-up kettlebell carry with the humerus at 90° of horizontal abduction and the forearm vertical to the floor (Fig. 1). After completing the first carry, subjects took a short break of around 30 s and then were told to complete a second kettlebell carry trial with the humerus at 45° of horizontal abduction and the forearm vertical to the floor (Fig. 2). The second group performed the same two carries, in reverse order.

Weight and distance for the kettlebell carries was standardized at 25 lbs for the kettlebell and 30 yards for the distance, with a turnaround at 15 yards. The 15 yard mark was clearly defined with tape, and subjects were instructed to walk at a comfortable pace. Subjects were instructed to maintain upright posture with shoulders parallel to the floor during the carries.

The 25 lbs was chosen as the standardized weight as the kettlebell carry is often performed as a recovery exercise in-gym for many of the athletes, and the individual athlete exercises discretion in choosing weights ranging from 15 lbs to 35 lbs; likewise, the 30 yards is the default standardized distance used during in-gym exercise.

Raw EMG signals were collected at 1,000 Hz by a Somaxis Cricket EMG sensor (Somaxis Inc., San Francisco, CA, USA). Data were sent in real time to an iPhone via Bluetooth and recorded by the Chirp for Cricket application (Somaxis Inc., San Francisco, CA, USA).

## Statistical analyses

The EMG sensor generated two columns of data for each pitcher, one containing the raw data and the second returning data from a specialized built-in cricket filter. For the sake of scientific test-re-test reliability and an open source protocol, the raw data was used. After disallowing a few aforementioned subjects, the remaining 33 subjects compromised 17 subjects who performed the 45° carry followed by the 90° carry (the first group) and 16 subjects who performed the 90° carry followed by the 45° carry (the second group).

As referenced above, surface EMG cannot and should not be used to implicate muscle activation; instead, EMG measures changes in the force-generation magnitude of the individual muscle fibers' membranes rather than the quantity of active muscle fibers, which is what generates muscle activation (*Vigotsky et al., 2018*). As such, EMG amplitude is more accurately a measure of muscle excitation. Even with these careful denotations, a number of filtering, normalizing, and smoothing measures were taken to clean the raw EMG data.

First, a band pass filter was used to remove components below and above selected frequency cut-offs of 10 Hz (the low end cutoff, traditionally used to remove electrical noise associated with wire sway, biological artifacts, etc.) and 450 Hz (the high end cutoff, used to eliminate tissue noise at the electrode site) (*De Luca et al., 2010*).

These cutoff frequencies were then applied with a Butterworth filter to transform each of the pitchers raw data into a filtered output, for better detection of the onset and offset of the kettlebell carries. A detection algorithm based on a double threshold scheme was applied to detect the start and end of the kettlebell carries in each pitchers'

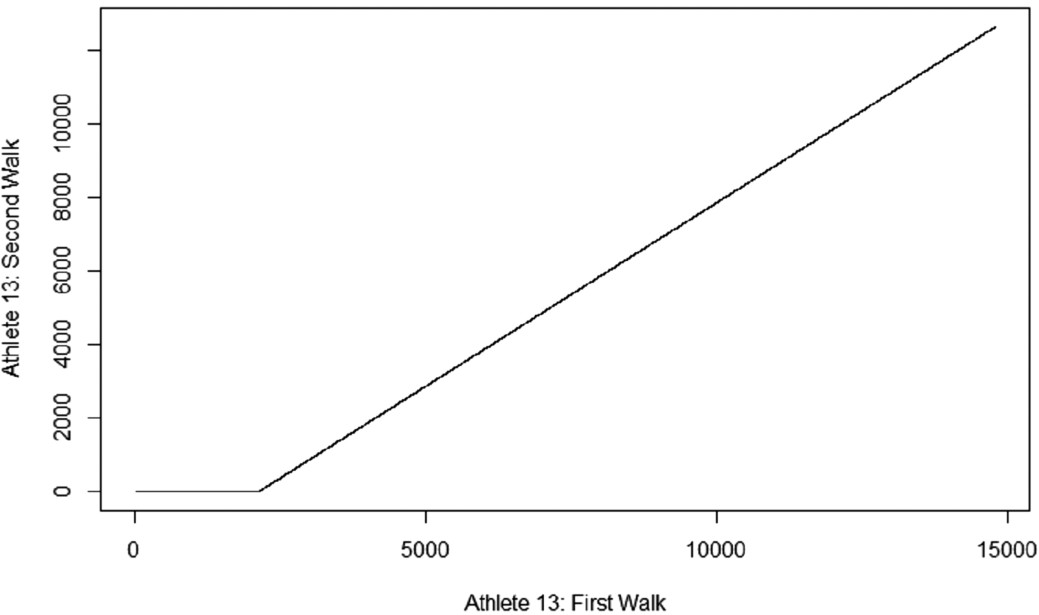

**Figure 6 Dynamic Time Warping example.** An example of Dynamic Time Warping as produced by R-Studio.               

continuous EMG data (*Bonato, D'Alessio & Knaflitz, 1998*). After these carries were identified, they were normalized with the peak MVIC value for each respective subject and smoothed with a moving-average based envelope of the normalized signal for further elimination of any phase shifts of the signal. Normalization of the signal, even post filtering, is required to enable a valid comparison between different individuals and different tasks (*Halaki & Ginn, 2012*). Per past EMG literature, the optimal window size of 200 was used for the said-MA smoothing (*Dieterich et al., 2017*).

Statistical analysis was done on both (a) intra-subject differences for the each of the 33 pairs of carries and (b) the group differences between all the 45° carries and all the 90° carries. Analysis revolved around using the mean frequencies and median frequencies of the EMG filtered signal because frequency-domain features have been shown to be the most ideal variables in identifying muscle fatigue and force (*Phinyomark et al., 2012*).

For the intra-subject comparisons, three different avenues of statistical comparisons were pursued: (a) the Effect Size Test (specifically, the Cohen's D estimate) as a method of statistical power analysis that looks at the standardized difference between means; (b) the Granger Test, which pinpoints predictive causality of the first carry towards the second one; and (c) the normalized distance from a dynamic time warping method that relies on minimizing the Euclidean distance off an optimally aligned time series (Fig. 6).

## RESULTS

Out of all 33 individuals, 22 had the larger mean frequency in the 45° kettlebell carry. Splitting the groups by the order of carries, the first group (90° carry performed first)

**Table 1 Looking at overall group differences.**

|  | Mean ($n = 33$) | SEM |
|---|---|---|
| 45° mean frequencies | 9.799 | 0.152 |
| 90° mean frequencies | 9.354 | 0.161 |
| 45° median frequencies | 0.071 | 0.003 |
| 90° median frequencies | 0.077 | 0.006 |

Note:
An analysis of the total group differences in the subject pool.

**Table 2 Group differences subset by order of walk.**

|  | Mean 1st group ($n = 17$) | SEM | Mean 2nd group ($n = 16$) | SEM |
|---|---|---|---|---|
| 45(°) walk MNF | 9.754 | 0.202 | 9.848 | 0.235 |
| 90(°) walk MNF | 9.202 | 0.203 | 9.516 | 0.253 |
| 45(°) walk MDF | 0.073 | 0.005 | 0.069 | 0.007 |
| 90(°) walk MDF | 0.083 | 0.007 | 0.072 | 0.010 |

Note:
Examining if the order of the posture changes made a significant difference. Photo credit: Alex Caravan.

found 10/16 subjects with a higher value in the 45° carry, while the second group found 12/17 subjects with the higher value in said 45° carry. Judging by the Cohen D statistics (where the widely accepted scales of significance are 0.2–0.5 *d* for a "small" effect, 0.5–0.8 *d* for a "medium" effect and anything above 0.8 *d* is deemed as a "large" effect), eight subjects had a large effect (seven of them with the higher mean in the 45° carry), eight subjects had a medium effect (four of them in the 45° carry), and 10 subjects had a small effect (six in the 45° carry) (*Sawilowsky, 2009*).

Looking at the Granger Test of forecasting probability (with a chosen alpha level of 0.05), the majority of the subjects had a significantly predictive first carry, with 10 of these 17 significant Granger estimates registering among the group that performed the 45° carry first.

Lastly, we looked at the normalized distance between the individuals where the Euclidean cost was minimized via the dynamic time warping alignment, as a method of verifying the Effect Size denoted by Cohen's D (Table 1). We found the "large" effects averaged a dissimilarity measure of 2.53, the medium effects averaged 1.60, and the small effects averaging around 1.00 (the differences that didn't register interestingly had around a 1.23 average).

Looking at the holistic groups of 45° and 90° carries, a paired *T* test was used to judge the significance of the difference between the two groups of mean frequencies (Table 2). The paired *T* test was chosen because the sample size was large enough to guarantee a normal distribution; in addition, the plot density and quantile-quantile plots all showed normality for both groups of means, while both groups comfortably failed to reject the Shapiro Test of normality (registering *p*-values of 0.879 and 0.791, where a sub 0.05 score would have been needed to reject the null hypothesis of normality). Furthermore, even the order-dependent subgroups of the means (the 45° and 90° carries of both the first and second group) fulfilled all the aforementioned normality checks.

The Paired *T* Test returned a significant value of 0.0180, with the mean frequencies of the groups being 9.799 and 9.354 for the 45° and 90° carries, respectively. The difference stems in larger part from the subjects who performed the 45° carry first, as a mean difference analysis of that subgroup returned a significant *p* level by itself (0.0325 versus a *p* level of 0.246 from the subgroup who performed the 45° carries second).

## DISCUSSION

The purpose of this investigation was to examine differences in the muscle engagement of the serratus anterior between 45° and 90° kettlebell carries. These differences were examined across both intra-subject carries and the overall groups of carry mean frequencies. While the intra-subject differences ranged from inconsequential differences to "large effect" differences, the grouped mean difference analysis returned a steady significant result of a 0.018 *p*-value.

As mentioned above, a number of considerations have to be taken in account in the emerging world of EMG analysis and statistical studies. Specifically, for the scope of this study the main concern resides in drawing inferences about how closely EMG amplitude indicates agonist muscle activation, as the raw signal of the EMG amplitude is often muddied by noise from the subject's unique muscle fiber recruitment and motor unit firing frequency (*Kuriki et al., 2012*), as well as the exact location of the EMG sensor and the quantity of the respective subcutaneous tissue on which it is placed (*De Luca, 1997*). Hence, the investigation adopted proper filtering (*Stegeman & Hermens, 2014*), normalization (*Alkner, Tesch & Berg, 2000*), and frequency extraction (*Phinyomark et al., 2012*).

There have been several studies concerning the EMG activity of the serratus anterior, although none have dealt with the exact nature of this study's investigation. A study examining variations in the EMG activity of different muscles throughout the push-up plus exercise found that SA activity was significantly greater during the "plus" phase of the exercise and was very malleable to different hand positioning and support surfaces (*Gioftsos et al., 2016*).

In addition, and perhaps more relevantly to this study's design, a 2003 study measured the EMG activity of separate trapezius muscles and the serratus anterior muscles across 10 different scapula-involved exercises (*Ekstrom, Donatelli & Soderberg, 2003*). It found that the highest EMG activations were achieved by, in order, the (a) Diagonal exercise with shoulder flexion, horizontal flexion, and external rotation and (b) the Shoulder abduction in the plane of the scapula above 120°, the two exercises most similar in movement to a 45° and 90° kettlebell carry, respectively. However, our findings were in contrary to the study, as we found the differences of the EMG amplitudes significant, while the aforementioned study found a nonsignificant difference. It is important to note, though, that the 2003 study involved submaximal voluntary dynamic contraction of the exercises versus the submaximal voluntary isometric contraction (SVIC) movements that were observed in this study.

Future studies could explore a number of things. The study could be reproduced before a workout rather than after, in case a pre-fatigue MVIC (and subsequent SVIC) generates different results. There could be further variations in the degree of the angle abduction of

the kettlebell carry: between 0° and 45° and between 45° and 90°. There could also be more variability in the weights used, to see if the differences change in magnitude with the weight used, or even reverses. Further, there could be investigations into prior postural deficiencies or upper body mobility and flexibility capabilities to try to control for differences in biomechanical advantages impacting just how much an individual relies on or stresses their serratus. A longitudinal study would be especially interesting in exploring how the serratus activation changes in behavior with the subject's own body and physical attributes.

## CONCLUSION

Our findings indicated that there is a significant increase in EMG amplitude for the serratus anterior in the 45° kettlebell carry versus the complementary 90° one for a population of adult male pitchers. For various reasons, it is hard to draw an exact match between muscle activation and EMG amplitude, but it appears likely that muscle activation of the serratus anterior is higher in the case of the former exercise. These findings can be used to increase the chances of athletes retaining arm health by offering 45° kettlebell carries as part of a thorough post-workout recovery exercise routine.

### Funding

The authors received no funding for this work.

### Competing Interests

Alex Caravan, John O. Scheffey, Sam J. Briend and Kyle J. Boddy are employed by Driveline Baseball Inc.

### Author Contributions

- Alex Caravan performed the experiments, analyzed the data, prepared figures and/or tables, authored or reviewed drafts of the paper, approved the final draft.
- John O. Scheffey performed the experiments.
- Sam J. Briend conceived and designed the experiments, performed the experiments.
- Kyle J. Boddy contributed reagents/materials/analysis tools, prepared figures and/or tables, authored or reviewed drafts of the paper, approved the final draft.

### Human Ethics

The following information was supplied relating to ethical approvals (i.e., approving body and any reference numbers):

Hummingbird IRB granted Ethical approval to carry out the study within Driveline Baseball's facilities—approval number 2017-58.

### Data Availability

The raw data are provided as Supplemental Files.

## Supplemental Information

Supplemental information for this article can be found online at http://dx.doi.org/10.7717/peerj.5044#supplemental-information.

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
