# Peer review of "Surface electromyographic analysis of differential effects in kettlebell carries for the serratus anterior muscles"

_PeerJ, doi:10.7717/peerj.5044_

## Round 0.1 · original submission · Minor Revisions

Based on the views of the three reviewers and I, I would like to invite you to revise your manuscript. Please be careful to take on board each of the three reviewers specific comments in terms of your revised manuscript.

·

Basic reporting

Line 28 - A reference here detailing how upward rotation maintains subacromial space and prevents rotator cuff impingement. A decrease in subacromial space and rotator cuff impingement may be caused by many factors including, but not exclusive to scapular upward rotation.
Line 38 - This is the first mention of the term "overhead stabilizers". It would be good to better define this earlier as it presents a little confusion for the reader. Previous mentions serratus anterior actions as upward rotation or scapular stabilization.
Lines 49-57 -These lines highlight the concerns/limitations of EMG. Potential limitations of measuring apparatus are typically seen in the Discussion section.
Line 47 -If there are any references highlighting the validity of EMG and serratus, it would be good to mention them here. If not the serratus specifically, then another scapular stabilizer perhaps?

Experimental design

Line 66 - Any exclusion criteria include past medical history (i.e. history of rotator cuff impingement, shoulder surgery, etc.) Any previous history may effect results.

Line 73 - How were the subjects randomized? If the study is to be replicated, then specifics on randomization should be included.

Line 91 - How long between each trial?

Line 97 - Why was 25 lbs of the kettlebell and 30 yards distance chosen?

Validity of the findings

Line 149 - Is the line supposed to read: 8 had a large effect vs. 8 had a large difference? The rest of the sentence discusses "effect".

Line 202 - There is confusion here. There is listed the 2003 as submaximal voluntary dynamic contraction (SVDC) of the exercises vs the submaximal voluntary dynamic contraction (SVIC) movements.-----I am unsure of the difference here? Possible grammatical error?

I always like a sentence or two referring to clinical implications of the study. Is there a clinical implication that can be drawn from this study?

Additional comments

Great work here! Thanks for the opportunity to review.

·

Basic reporting

This article is, on the whole, well written in an understandable manner; however, minor proofreading and revision may be needed at lines 29-31, 208-211.

Regarding potential influences on EMG activity readings, I feel that a reference should be cited at the end of lines 48-52; however, if the Kuriki article cited in line 53 addresses these concerns, a simple re-wording of the sentence would suffice.

Experimental design

Well designed experiment. For clarification, what is the rationale for performing the protocol post-throwing? Would discussion of potential decrease in MVIC due to potential fatigue be warranted, as well as directing potential future investigation to include assessing throwing athletes in a "fresh" state?

Regarding protocol, I believe it would be beneficial to include the specified rest time prior to performing each trial; while neuromuscular fatigue is not uniform across the population, this would give some indicator of potential fatigue influencing repeated trials for the study design.

Validity of the findings

Data appears to be well collected and analyzed.

Regarding the conclusion, I feel that it may be necessary to narrow the scope of the conclusion statement, regarding adult male throwers.

·

Basic reporting

Abstract:

Line 12: write out electromyography (EMG) the first time you use it in abstract
line 14: change "roughly college age" to "men ages xx-xx"
line 14: change to "professional baseball pitchers";
line 16: remove "after a brief break" since you explain that in next sentence.
lines 21-25: for clarity, i would include "EMG amplitude" or "EMG signal" into those sentences when talking about the results.

Introduction:

Overall well written and succinct. Nicely illustrates the background and reason for the study. I would recommend moving the purpose followed by the hypothesis to the end of the introduction.

Line 32: do people specifically use it for “recovery”? I would consider removing that word if possible. (if recovery is removed consider modification to line 46 as well)
Line 33-34: Change to “ When performed with correct technique, kettlebell carries…”
Line 35: remove “often”
Line 42: I would refer to figure 4 and 5 to illustrate the positions. Figures will need to be renumbered if this change is made.

Methods:

I would move line 69-70 on IRB approval to first sentence of this section. move consent info found in lines 67-69 to second sentence.
Please use consistent horizontal shoulder abduction angles in relation to each other. Try to clarify what zero degrees is. At times 45 is used but at other times 90 degrees and 125 degrees. I typically think of the arm forward flexed to be zero degree and figure 5 to be 90 degrees.
Line 60: add city,state of driveline baseball
Line 72: change “ask” to “recruited”
Line 73: discuss method of randomization
Line 87-88: please clarify the description in figure 2 and 3 to better reflect what is occurring in the picture

Statistical analysis

Nice work explaining this section and how filters are applied.

Results

Clearly written.
Line 171-175 – I would recommend 3 significant digits for P values

Discussion

Nice opening paragraph restating purpose and your most important findings.
Line 196 – change “this study” to “it” as it appears you are referring to the ekstrom study
line 211-213 – I would reword or remove “especially pertinent in nature because of fickle behavior of EMG

Experimental design

Well designed project with purpose and method clearly stated. Study would be easy to replicate because of detail within this paper.

Validity of the findings

This is an exercise that is widely done but the exact arm position to activate the serratus anterior has not been well established previously. These findings begin to help the understanding of what muscles are being trained with this kettlebell exercise.

Additional comments

Well done study! Excited to see more of this type of research from your research group!

---

## Round 0.2 · accepted · Accept

Congratulations, you have addressed all comments of the reviewers and myself

·

Basic reporting

No Comment

Experimental design

No Comment

Validity of the findings

No Comment

Additional comments

Great work! Looking forward to more follow-ups on this topic.

·

Basic reporting

Well done. Rephrasing this section addressed previous concerns.

Experimental design

Concerns were addressed well in this section. Regarding line 116, phrasing "around" as "approximately" or "as near to 30 seconds as possible" may aid future investigators in repeating the procedures by limiting ambiguity.

Validity of the findings

Well done study.

Additional comments

Thank you for the opportunity to review this manuscript and I look forward to seeing more data from your group in the future.

·

Basic reporting

all my concerns were addressed. looks great!

Experimental design

all my concerns were addressed. looks great!

Validity of the findings

all my concerns were addressed. looks great!

Additional comments

all my concerns were addressed. looks great!